# Evaluation of Biochemical Properties, Antioxidant Activities and Phenolic Content of Two Wild-Grown Berberis Fruits: *Berberis nummularia* and *Berberis*
*atrocarpa*

**DOI:** 10.3390/foods11172569

**Published:** 2022-08-25

**Authors:** Buhailiqiemu Abudureheman, Xinyue Zhou, Xipan Shu, Ziqi Chai, Yongping Xu, Shuying Li, Jinhu Tian, Haibo Pan, Xingqian Ye

**Affiliations:** 1Xinjiang Institute of Technology, College of Food Science and Engineering, Aksu 843000, China; 2College of Biosystems Engineering and Food Science, National-Local Joint Engineering Laboratory of Intelligent Food Technology and Equipment, Zhejiang Key Laboratory for Agro-Food Processing, Zhejiang Engineering Laboratory of Food Technology and Equipment, Fuli Institute of Food Science, Zhejiang University, Hangzhou 310058, China; 3Postdoctoral Workstation of Dalian SEM Bio-Engineering Technology Co., Ltd., Dalian 116620, China

**Keywords:** *Berberis nummularia*, *B. atrocarpa*, biochemical characterization, antioxidant activity, total phenols content, total phenolic content

## Abstract

To evaluate the potential health-promoting benefits of *Berberis nummularia* and *B. atrocarpa* fruits, the biochemical properties (nutrition component, mineral substance, organic acids), total phenolic and flavonoid content and antioxidant (DPPH, FRAP, ABTS and ORAC) capacity of ethanol extracts of *B. nummularia* and *B. atrocarpa* fruits wild-grown in Xinjiang were analyzed. The results indicated that there were no meaningful differences (*p* > 0.05) between the ash (1 ± 0.1 and 1 ± 0.0 g/100 g), fiber (16 ± 1.0 and 18 ± 1.4) and carbohydrate (57 ± 1.8 and 56 ± 1.8 g/100 g) content, respectively, in the dry fruits of *B. nummularia* and *B. atrocarpa*. The total fat (7 ± 0.4 and 5 ± 0.1 mg/100 g), soluble sugar (23 ± 0.6 and 12 ± 1.4 g/100 g), titratable acidity (18 ± 2.5% and 14 ± 1.3%) content, and energy value (330.86 and 314.41 kcal/100 g) of *B. nummularia* was significantly higher than that of *B. atrocarpa* fruits. Both species contain malic acid, acetic acid, tartaric acid, citric acid and fumaric acid, in which, malic acid is the dominant organic acid. The organic acid and mineral components of *B. nummularia* fruits were significantly higher than that of *B. atrocarpa* (*p* < 0.05). The total phenolic and flavonoid content of *B. nummularia* were 2 ± 0.0 mg GA/g DW and 2 ± 0.0 mg RE/g DW, respectively, which were significantly lower than the total phenolic and flavonoid content of *B. atrocarpa* (12 ± 0.1 mg GA/g DW and 9 ± 0.0 mg RE/g DW). The antioxidant capacity of *B. nummularia* (4 ± 0.1 mg Ascorbic acid/g DW for DPPH, 32 ± 0.1 mg Trolox/g DW for FRAP, 80 ± 3.0 mg Trolox/g DW for ABTS and 60 ± 3.6 mg Trolox/g for ORAC was significantly lower than that of *B. atrocarpa* (12 ± 0.0 mg Ascorbic acid/g DW for DPPH, 645 ± 1.1 mg Trolox/g DW for FRAP, 304 ± 3.0 mg Trolox/g DW for ABTS and 155 ± 2.8 mg Trolox/g for ORAC). *B. atrocarpa* fruits showed significantly higher antioxidant capacity than that of *B. nummularia*. The fruits of the two species can be used in food coloring and nutritional supplements, and consumption of the fruits can aid in weight control and reduce blood glucose or cholesterol.

## 1. Introduction

The nutritional content and medicinal value of wild edible fruits are becoming increasing valuable due to a global trend in consuming natural foods with pharmacological properties [1]. Numerous studies have reported berberis to be a rich source of nutrients and phytocompounds and exhibit antioxidant properties [1,2]. The fruits of the wild species murta (*Ugni molinae* Turcz) and calafate (*Berberis buxifolia* Lam.), both of which grow natively in Patagonia, can be sources of anthocyanins, and bear an ascorbic acid content of 210 mg/100 g F.W. [3] and 74.0 mg/100 g FW, respectively [4]. The content of lipids and crude protein in *Hippophaë rhamnoides* (sea buckthorn) was 21 ± 0.1% and 17 ± 0.4% DWP [5]. The antioxidant capacity of ethanol extracts of *H. rhamnoides* fruits were measured as 294 ± 6.5 µM TE/g (ORAC), 269 ± 7.1 µM TE/g (ABTS), 102 ± 4.3 (DPPH). *H. rhamnoides* fruits contain a rich amount of phenolic compounds such as proanthocyanidins, gallocatechins and flavonol glycosides [5]. Gıdık (2021) [1] showed that among the cultured and hybrid plants of *B. crataegina* DC., *B. integerrima* Bunge, and wild *B. vulgaris* L., the last one had the highest FRAP (621.02 ± 25 µmol/g) values. The phenolic substances content of the wild Saskatoon berry (*Amelanchier alnifolia* Nutt.) was 3.80 g GA/kg F.W. and the antioxidant capacity reached 5.05 g of ascorbic acid equivalent (ABTS) [6]. Among antioxidants, phenolic compounds such as flavonoids, phenolic acids and anthocyanins have antitumor, antidiabetic and immunomodulatory effects [7,8,9,10,11]. Therefore, wild edible berberis grown in different areas of the world and their extracts are widely used as a source of natural antioxidants in food and pharmaceutical industries. With these benefits, there is an increasing demand for different products derived from the berberis fruit, and thus the nutritional composition, biological activities and various potentials of wild edible berberis fruits should be determined [8].

Northwest China is one of the main sources of genus Berberis. *Berberis nummularia* Bunge and *B. atrocarpa* Schneid. are the two wild species distributed on valleys at altitudes of 550–1100 m, in Xinjiang, northwest China [12]. The fruit of *B. nummularia* and *B. atrocarpa* are oval- and ovoid-shaped [13], and are culturally important species owing to their traditional uses in medicine. Moreover, the fresh fruits have been used as an ingredient in making juices and sauces [2,12,13]. Some research has been conducted on the roots and fruit of *B. nummularia* and *B. atrocarpa,* for medical purposes. Berberine, Vitamin C and flavonoids have been isolated from the fruit and roots of *B. nummularia* and *B. atrocarpa* [8,14]. However, there are few reported research materials regarding the biochemical composition and antioxidant activity of the fruits of these two species in northwest China. Therefore, the aim of this study was to assess the biochemical characterization, and evaluate the antioxidant activities of the wild edible fruits of *B. nummularia* and *B. atrocarpa* for the development of functional food and nutraceuticals.

In this study, the nutrient component, mineral substance and organic acid compositions of the fruits of wild *Berberis nummularia* Bunge and *Berberis atrocarpa* Schneid. were determined. With this, the total flavonoid and phenolic content, and antioxidant (DPPH, FRAP, ABTS and ORAC) capacities were determined in order to gain insight regarding the bioactive content of the fruits of *B. nummularia* and *B. atrocarpa*.

## 2. Materials and Methods

### 2.1. Materials and Chemicals

Fruits of *B. nummularia* and *B. atrocarpa* obtained from 30 different trees were collected from natural habitat in Aksu (XJBI 10020070) (41°29′85″ N, 79°98′14″ E) and Yili (KUN 1448344) (43°32′55″ N, 84°03′46″ E), Xinjiang, respectively, in October 2021 (Figure 1) and identified by Professor Duan, S.M., from the Xinjiang Institute of Ecology and Geography, Chinese Academy of Sciences. The fruits were dried at room temperature and then frozen at −80 °C until further analysis.

The organic acid standards (tartaric acid, malic acid, acetic acid, citric acid and fumaric acid) were purchased from Sigma-Aldrich (Burlington, MA, USA). Gallic acid, Trolox and ascorbic acid were obtained from Yuanye Biotechnology Ltd. (Shanghai, China). Ethanol and other chemicals of analytic grade were purchased from Sinopharm Chemical Reagent Co. (Shanghai, China).

### 2.2. Proximate Composition Analysis of B. nummularia and B. atrocarpa Fruits

The incineration of 3 g of each sample into ash was performed using a muffle furnace at 600 °C for 6 h. Crude fat was extracted with n-hexane in a Soxhlet extractor. Total dietary fibre content was determined using the Megazyme International total dietary fibre assay (adopted from AOAC method 985.29) [15]. The total soluble sugar was determined according to Tang et al. (2021) [16] and expressed as g/100 g on a dry weight basis. The soluble protein content was determined according to Bradford (1976) [17] and expressed as g/100 g on a dry weight basis. Titratable acidity (TTA) was determined according to AOAC 942.15. Energy was expressed as kcal/100 g, using the following formula:(1)Energy[kcal/100g]=4×(protein+carbohydrate)+9×fat+2×fiber

The total soluble sugar components were analyzed by HPLC (Agilent 1200) equipped with a refractive index detector (RID-10) (model 1200 series), a Rheodyne 7725 injector of 20 μL loop volume and a column (120 Å, 250 mm × 4.6 mm, 3 um (Athena NH2)) maintained at 40 ℃. The mobile phase was a mixture of DI acetonitrile: water = (75:25 *v*/*v*). The flow rate was kept at 1.0 mL/min. This method was adapted and revised from Bouhlali et al. (2017) [18]. A calibration curve was prepared using fructose and glucose standard solution (0.130625, 0.26125, 0.5225, 1.045 and 2.09 mg/mL with R^2^ = 0.9984) and (0.13375, 0.2675, 0.535, 1.07 and 1.605 mg/mL with R^2^ = 0.9965), respectively.

### 2.3. Determination of Organic Acids

The organic acids were extracted according to Wu et al. (2022) [19]. In total, 5 g each of *B. nummularia* and *B. atrocarpa* fruit powder were mixed with 25 mL water, and then placed into an ultrasonic bath for 60 min continuously at 80 °C at a frequency of 40 kHz. The solution was further centrifuged at 9050× *g*/min at 20 °C for 15 min. The supernatants were collected, then filtered by 0.22 µm membrane (Millipore Millex-HV Hydrophilic PVDF, Millipore, Burlington, MA, USA). The organic acids were detected by an HPLC (Agilent 1260, Agilent Technologies, Santa Clara, CA, USA) equipped with variable wavelength vacuum chromatograph and high-efficiency UV detector. The system was equipped with a C18 AQ (4.6 mm × 250 mm, 5 μm) and DAD detector. The mobile phase consisted of 1% phosphoric acid solution (97.5%) and methanol (2.5%), the flow rate was 0.5 mL/min, and the temperature was 35 °C. The injection volume was 10 μL, and the run time was 15 min. A wavelength of 210 nm was employed to detect the organic acids. An external mixed standard of tartaric acid, malic acid, acetic acid, citric acid and fumaric acid was prepared at a concentration of 0–1 mg/mL (Figure 2).

### 2.4. Mineral Content Analysis

The analysis of mineral content (Ca, K, Na, Mg, P, Zn, Cd, Fe, Cu, Mn) was carried out according to Kookal and Thimmaiah (2018) [20]. A total of 0.5 g of the powdered sample was placed into a 50 mL conical flask, soaked in 10 mL of diacid mixture (concentrated HNO_3_ and perchloric acid: 5:1, *v:v*) and kept overnight. Then, the samples were digested by heating till the solution became transparent. After cooling, final volume was made up to 50 mL with 0.1 mol/L HNO_3_ and used for the determination of individual minerals after being filtered by a 0.22 µm membrane (Millipore Millex-HV Hydrophilic PVDF, Millipore, Burlington, MA, USA) with the help of the ICP-MS device at Zhejiang University.

### 2.5. Extraction Process for Total Phenolic Content (TPC), Total Flavonoids Content (TFC) and Antioxidant Capacity

The 5 g samples of *B. nummularia* and *B. atrocarpa* fruits were mixed with 50 mL of 80% ethanol extract. Subsequently, they were sonicated for 60 min at 70 °C at a frequency of 40 kHz. Then, the solutions were centrifuged at 5000× *g*, 20 °C for 15 min. The supernatants were filtered, and dried using a rotary evaporator at 50 °C followed by freeze-drying at −52 °C for 48 h. The dried extracts were reconstituted in 10 mL 80% ethanol and kept at 4 °C prior to analysis of the total phenolic and flavonoids content, and antioxidant capacity.

#### 2.5.1. Total Phenolic Content (TPC)

Total Phenolic Content (TPC) was determined according to the modified Folin–Ciocalteu colorimetric method at 760 nm [21], by using gallic acid as a standard. Results were expressed as mg gallic acid equivalents/g dry weight (DW) of sample.

#### 2.5.2. Total Flavonoids Content (TFC)

The aluminum chloride colorimetric method described by Pu et al. (2018) [21] was modified to determine the total flavonoids content (TFC), at 420 nm, by using rutin as a standard. Results were expressed as mg rutin equivalents/g (DW) of sample.

### 2.6. Antioxidant Activity

#### 2.6.1. 1,1-Diphenyl-2-picrylhydrazyl (DPPH) Radical Scavenging Activity

The DPPH assay was performed using the method of Tian et al. (2016) [22] with some modifications at 510 nm, by using ascorbic acid as standard. Results were expressed as mg ascorbic acid/g DW.

#### 2.6.2. The Ferric Reducing Antioxidant Power (FRAP) Assay

The FRAP assay was performed using the method of Tian et al. (2016) [22], at 593 nm, by using Trolox as a standard. Results were expressed as mg Trolox/g DW.

#### 2.6.3. Assay of ABTS Radical Scavenging Activity

The ABTS assay was performed using the method of Fang et al. (2009) [23] with some modifications at 734 nm, by using Trolox as a standard. Results were expressed as mg Trolox/g DW.

#### 2.6.4. Measurement of Oxygen Radical Absorbance Capacity (ORAC)

The ORAC assay was performed using the method of Liu et al. (2016) [24] with some modifications, with excitation at 485 nm and emission at 538 nm, once every 2 min, for 2 h, by using Trolox as a standard. The area under the fluorescence curve (AUC) was calculated for each well and the AUC of the blank well was subtracted, to obtain the net AUC. The ORAC value for each sample was calculated from the Formula (2), where CT is the time interval of 2 min.
(2)$$\mathrm{AUC}=\frac{0.5\times f_{1}+f_{2}+f_{3}+\ldots +f_{i}+\ldots +f_{58}+f_{59}+f_{60}}{f_{i}}\times \mathrm{CT}$$

The result was expressed as mg Trolox/g DW.

### 2.7. Statistical Analysis

Statistical analysis of this study was performed using the SPSS (version 18.0) software (IBM, Armonk, NY, USA). All results were expressed as means ± standard error. One-way ANOVA was used to analyze the difference between nutritional components, mineral contents, organic acids, total phenolic and flavonoid contents and antioxidant (DPPH, FRAP, ABTS and ORAC) capacity of ethanol extracts of samples at *p* < 0.05. Figures were drawn using Sigma plot version 12.0.

## 3. Results and Discussion

### 3.1. Composition of Fruits of B. nummularia and B. atrocarpa

The composition including the fat, protein, fiber, ash, soluble sugar (fructose and glucose), titratable acid, carbohydrates, and energy value of berberis is shown in Table 1.

The nutrient contents of the analyzed samples (protein, fat, sugar, titratable acid, carbohydrates and energy value) were significantly different between the *B. nummularia* and *B. atrocarpa* fruits (*p* < 0.05), except for fiber and ash. The protein content represents one of the important indicators in evaluating fruit quality and nutritional value [25]. The soluble protein content of *B. nummularia* (3 ± 0.1 g/100 g) was significantly lower than that of *B. atrocarpa* (4 ± 0.1 g/100 g) (*p* < 0.05). The soluble protein content of the two berberis was higher than that of the fruits of *B. microphylla* [10], in which, the soluble protein content was only 1 ± 0.1%. However, the protein content of the two species were lower than that in *Hippophaë rhamnoides* (sea buckthorn), in which, the protein content was 17 ± 0.4% DWP [5]. Thus, these fruits are shown to be a good source of protein for the human diet. The contents of crude fat, soluble sugar, titratable acid and crude fat in *B. nummularia* were significantly higher than those in *B. atrocarpa* (*p* < 0.05). The total amount of soluble sugar contained in *B. nummularia* (23 ± 0.6 g/100 g) is twice as high than that of *B. atrocarpa* (12 ± 1.5 g/100 g). The HPLC-RID chromatograms showed that the *B. nummularia* appears to have a significantly higher content of fructose and glucose, sharing this result with *B. vulgaris* [26].

Dietary fiber is an essential aspect of good nutrition. The fiber content measured up to 16% for both *B. nummularia* and *B. atrocarpa*, which is in line with previous reports of other berberis fruit [27]. Researchers suggest consumption of the dried-fruit, which are rich in dietary fiber, is beneficial for weight control and reducing blood glucose [28] due to increased satiety and intestinal regulation [29]. The content of carbohydrates was about 56% for the two fruits and may be a result of the high dietary-fiber content. The measured energy values were 330.86 kcal for *B. nummularia* and 314.41 kcal/100 g for *B. atrocarpa*.

### 3.2. Mineral Contents

The mineral contents and their recommended daily allowances at nutrient reference values (NRVs), are presented in Table 2. Kookal and Thimmaiah (2018) [20] reported that, potassium (K), calcium (Ca), magnesium (Mg) and iron (Fe) are especially important because they are often deficient in different groups. In this study, we observe that the distribution of the total amount of minerals contained in the two fruits is slightly different (*p* < 0.05). Among the minerals, potassium was found to comprise the highest concentration in the fruits of the two species, measured at 1380 ± 13.6 mg/100 g for *B. nummularia* and 1225 ± 10.6 for *B. atrocarpa*, which constitutes 70.24% and 79.56% of the total mineral content, respectively. This is line with a previous study that reported potassium as the most abundant mineral element in wild fruits [26]. The content of calcium found in *B. nummularia* (160 ± 16.7 mg/100 g) was higher than that of *B. atrocarpa* (132 ± 11.7 mg/100 g). The portion of calcium would cover up to 21% of NRV for this nutrient. Magnesium was also an abundant mineral in *B. nummularia* and *B. atrocarpa*, respectively, delivering 109 ± 1.5 mg/100 g and 58 ± 0.9 mg/100 g, which corresponds to up to 13% of NRV. The mineral contents in the two fruits of the present study were comparable to the mineral composition of three Iranian berberis species published by Rahimi-Madiseh et al. (2016) [30]. The content of magnesium in three Iranian samples ranged from 197 ± 16.0 to 541 ± 34.0 mg/100 g, while in two Turkish samples, namely *Berberis vulgaris* L. and *B.*
*crataegina* DC was slightly lower, ranging from 85.07 to 97.95 mg/100 g [26]. Zn and Mn are present in very small amounts in the two species. *B. atrocarpa* fruit has low levels (10 ± 0.3 mg/100 g) of iron (Fe) and Cu (1 ± 0.0 mg/100 g) content, but not present in *B. nummularia*. Noticeably, the content of Na in *B. nummularia* was significantly higher than in *B. atrocarpa* and other berberis [2,27]. This is related to the growing conditions of this species. Chen et al. (2017) [31] reported that barberry plants preferentially uptake Na when grown in elevated salinity conditions, which is consistent with our result as *B. nummularia* bears resistance to high salinity conditions in its living environment [32].

### 3.3. The Organic Acids Contents

Organic acids are used as preservatives, antioxidants and acidulants [33], which influence fruit coloration and ripeness. Organic acids are different in different fruits, as citrus [34] and wild apples [35] mainly contain citric acid, while cultivated apples [36] and pears [37] predominantly contain malic acid. Glew et al. (2003) [38] showed the fruits of a wild species called medlar (*Mespilus germanica* L.) contain around 4 g/kg F.W. of citric acid and malic acid. Our results showed the presence of tartaric acid, malic acid, citric acid, acetic acid and fumaric acid both in *B. nummularia* and *B. atrocarpa* fruits (Figure 2). The organic acid content ranged from 0.48 to 118.8 mg/g. There were significant differences (*p* < 0.001) in the content of total organic acid between the two fruits (Figure 2). Malic acid was the main organic acid for the two fruits, and was significantly higher in *B. nummularia* (119 ± 0.1 mg/g) than in *B. atrocarpa* (84 ± 0.1 mg/g). The content of citric acid and tartaric acid varied from 0.48 to 4.23 mg/g. The fumaric acid was present at low concentrations in both two fruits. The difference in the organic acid contents of the two species is likely due to genetic effects.

### 3.4. Total Phenolic Content (TPC) and Total Flavonoids Content (TFC)

The TPC of the crude extracts of *B. nummularia* and *B. atrocarpa* fruits were 2 ± 0.0 mg GA/g and 12 ± 0.1 mg GA/g DW, respectively, which is significantly different from each other (*p* < 0.05) (Figure 3). The total amount of phenol obtained from the two species in this study is less than those obtained in the results of Aliakbarlu et al. (2018) [39] in which, the highest total phenolic content of *B. vulgaris* was 92.75 GA mg/g. However, the total phenolic content of *B. atrocarpa* is more than most of the studies reported previously. Rybicka et al. (2021) [27] showed that the total phenolic content of Goji, Chokeberry, Physalis and Juniper were less than 30.12 mg GAE/g. Okatan and Çolak (2018) [40] obtained the total phenol contents between 11.98 and 26.17 mg GA/g FW in sixteen barberry genotypes in Turkey. Boeri et al. (2020) [10] reported the total phenol contents of *B. microphylla* fruits were 1035.03 mg GA/100 g FW. The previous studies showed that the environmental and nutrient conditions of the plant’s habitat, and the genotype and ripening steps of the fruits affect the formation and accumulation of secondary metabolites [41,42]. *B. nummularia* and *B. atrocarpa* fruits in this work were collected from a waste gravel and ravine in Aksu and Yili with semi-arid climates. This could explain the higher TPC determined in this work with respect to Ruiz et al. (2014) [41] who used samples from hyper-humid to humid climates (south of Chile).

Different amounts of TFC were revealed in the fruits of two berberis (Figure 3). Hassanpour and Alizadeh (2016) [43] reported TFC of berberis fruits ranging from 1.3 to 2.8 mg/g FW. Zovko et al. (2010) [44] reported TFC of berberis ranging from 0.12 to 4.23 mg/g. Rybicka et al. (2021) [27] also reported Chokeberry and Juniper with total phenolic contents of 7.75 mg/g and 3.23 mg/g DW. However, they measured 13.25 mg/g DW in Physalis, which was higher than the results of our study. In our study, the TFC of *B. atrocarpa* fruits (9 ± 0.0 mg RE/g DW) accounted for 72% of the total content of phenols. The results of TFC content in *B. atrocarpa* fruits are in line with the literature reporting flavonoids as the predominant phenolic group in plant foods accounting for nearly 75% of the dietary phenols [45]. For the fruits of *B. nummularia*, it was revealed that the TFC exceeded the content of phenols.

### 3.5. Antioxidant Activity of B. nummularia and B. atrocarpa Fruit Extracts

Evaluating the antioxidant capacity of natural resources is important in the modern health lifestyle context. The highest antioxidant capacity of fruits stemmed mainly from phenolic compounds [46]. However, the type of extraction solvent used significantly influenced the phytochemical composition and biological activity of the fruits [47]. Wang et al. (2022) [47] studied the relationship between phytochemical content and the biological activities of noni (*Morinda Citrifolia* L.) fruit extracts (NFEs) and the results showed that the total phenolic content in Bet-Gly (Betaine: Glycerol) extracts was highest at 11.89 mg GAE/g DW, but the ethanol extracts exhibited the strongest antioxidant activities, with the results from DPPH, ABTS+, FRAP and reducing power values measuring 61.31, 123.46, 347.37 and 312.42 μmol TE/g DW, respectively. In the present study, the total antioxidant activity of the ethanol extract of *B. nummularia* and *B. atrocarpa* fruits ranged from 4 ± 0.1 to 12 ± 0.0 mg ascorbic acid/g DW for DPPH, from 32 ± 0.1 to 645 ± 1.1 mg Trolox/g DW for FRAP, from 80 ± 3.0 to 304 ± 3.0 mg Trolox/g DW for ABTS and from 61 ± 3.6 to 155 ± 2.8 mg Trolox/g DW for ORAC assays, respectively. This is consistent with the results demonstrating significantly higher values in TFC and TPC of the four assays of *B. atrocarpa* than that of *B. nummularia* (*p* < 0.05) (Figure 4). According to previous research, higher values indicate better antioxidant ability of the extracts [45]. In this study, *B. atrocarpa* fruit demonstrated a significantly higher antioxidant capacity in the four antioxidant assays than that for *B. nummularia*. Gholizadeh-Moghadam et al. (2019) [42] reported that *B. vulgaris* and *B. crataegina* had higher antioxidant capacities than *B. integerrima*, which is in agreement with our present results.

## 4. Conclusions

Wild and medicinal plant species have been used for many purposes. It is important to determine the biochemical properties and antioxidant capacity of these plants when intended for consumption. In this paper, we approximate the traditional knowledge of *B. nummularia* and *B. atrocarpa* fruits with the scientific research on both species for the first time. Although the nutritional content of *B. nummularia* was higher than *B. atrocarpa,* the fruits of both species appear to be a good source of potassium and calcium, malic acid and crude fiber. It was found that the TPC, TFC, as well as the potential antioxidant properties of *B. atrocarpa* demonstrate significant predominance.

The biological activity of phytochemicals indicate that *B. nummularia* and *B. atrocarpa* are also a promising source of natural antioxidants, with nutraceutical potential. Despite their high nutritional properties, the fruits of *B. nummularia* and *B. atrocarpa* are not considered enough and not properly recognized. When the important biochemical components are considered, one can deduce that the fruits of *B. nummularia* and *B. atrocarpa* are suitable for use in the food industry as food coloring, nutritional supplements, in addition to health and medical applications regarding weight control and blood glucose or cholesterol reduction.

## Figures and Tables

**Figure 1 foods-11-02569-f001:**
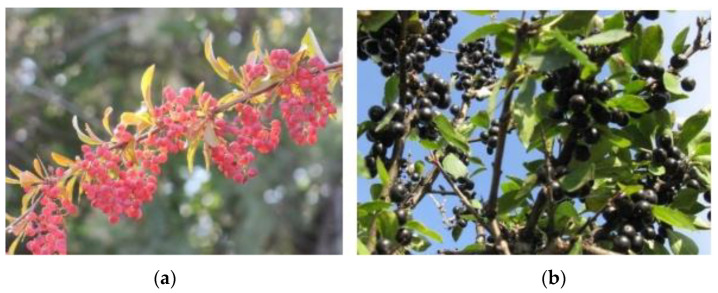
The fruits of *B. nummularia* and *B. atrocarpa*. (**a**) *B. nummularia*. (**b**) *B. atrocarpa*.

**Figure 2 foods-11-02569-f002:**
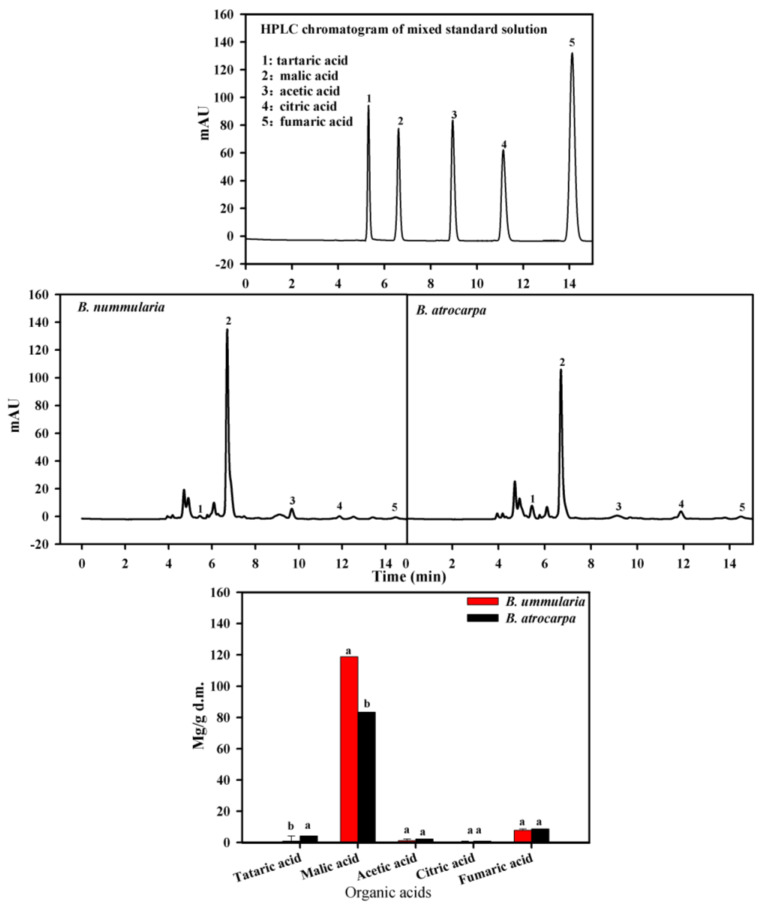
HPLC chromatogram of mixed standard solution and organic acid content in the fruits of *B. nummularia* and *B. atrocarpa.* Mean ± SD indicate three replicates. Different letters indicate significant difference (*p* < 0.05).

**Figure 3 foods-11-02569-f003:**
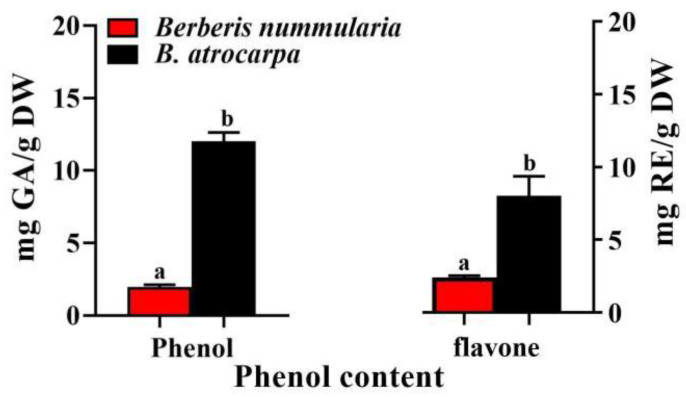
Total phenolic content (TPC), total flavonoid content (TFC) of two *B. nummularia* and *B. atrocarpa* fruits. Mean ± SD indicate three replicates. Different letters indicate significant difference (*p* < 0.05).

**Figure 4 foods-11-02569-f004:**
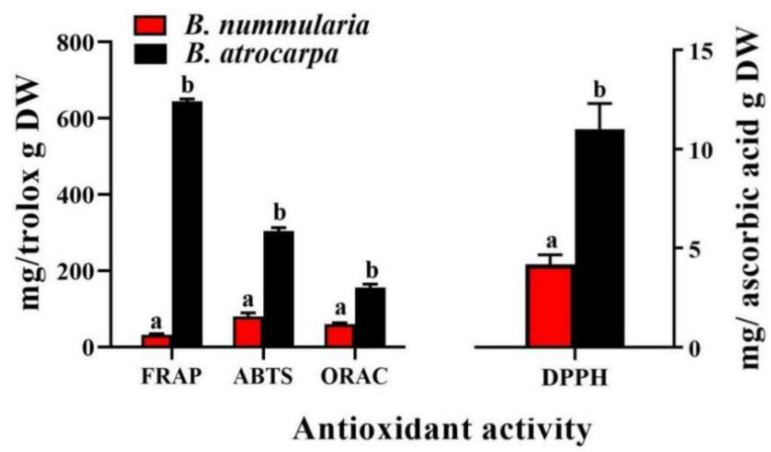
The antioxidant activity of *B. nummularia* and *B. atrocarpa* fruits. Mean ± SD indicate three replicates. Different letters indicate significant difference (*p* < 0.05).

**Table 1 foods-11-02569-t001:** Nutritional constituents of *B. nummularia* and *B. atrocarpa* fruits.

Fruits	*B. nummularia*	*B. atrocarpa*
Protein [g/100 g d.m. ^1^]	3 ± 0.1 ^a^	4 ± 0.1 ^b^
Fat [g/100 g d.m. ^1^]	7 ± 0.4 ^a^	5 ± 0.1 ^b^
Fiber [g/100 g d.m. ^2^]	16 ± 1.0 ^a^	18 ± 1.4 ^a^
Ash [g/100 g d.m. ^1^]	1 ± 0.1 ^a^	1 ± 0.0 ^a^
Soluble sugar [g/100 g d.m. ^1^]	23 ± 0.6 ^a^	12 ± 1.5 ^b^
Fructose (mg/g d.m. ^1^)	36 ± 0.2 ^a^	6 ± 0.1 ^b^
Glucose (mg/g d.m. ^1^)	29 ± 0.1 ^a^	10 ± 0.0 ^b^
Titratable acid (mg/g d.m. ^1^)	18 ± 2.5 ^a^	14 ± 1.3 ^b^
Carbohydrates [g/100 g d.m. ^2^]	57 ± 1.8 ^a^	56 ± 1.8 ^a^
Energy value [Kcal/100 g d.m. ^3^]	330.86 ^a^	314.41 ^b^

^1^ d.m. dry matter. ^2^ The carbohydrate content was calculated by subtracting fat, total fiber, ash, protein and moisture content from 100%. Mean ± SD indicate three replicates. ^3^ Energy value was calculated based on protein, fat and carbohydrate content. Superscripts of different letters within a row indicate significant differences (One-Way ANOVA, *p* < 0.05).

**Table 2 foods-11-02569-t002:** The contents of elements in the fruits of two Berberis species.

Fruits	Macroelements [mg/100 g d.m. ^1^]						Microelements [mg/100 g d.m. ^1^]			
	Ca	K	Na	P	Mg	Zn	Cd	Fe	Cu	Mn
^2^ NRV	800	2000	-	700	375	15	-	15	1	2
*B. nummularia*	160 ± 16.7 ^a^	1380 ± 13.6 ^a^	198 ± 16.6 ^a^	109 ± 4.3 ^a^	109± 1.5 ^a^	6.4 ± 0.4 ^a^	0	0 ^b^	0 ^b^	1 ± 0.0 ^a^
*B. atrocarpa*	132 ± 11.7 ^b^	1225 ± 10.6 ^b^	18 ± 1.4 ^b^	93 ± 1.3 ^b^	58± 0.9 ^b^	2 ± 0.3 ^b^	0	10 ± 0.3 ^a^	1 ± 0.0 ^a^	1 ± 0.0 ^a^

^1^ d.m. dry matter. ^2^ NRV Nutrient Reference Value. Superscripts of different letters in each column indicate significant difference (*p* < 0.05).

## Data Availability

The data presented in this study are available on request from the corresponding author. The data are not publicly available due to privacy reasons.

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
