# Peer review of "Evaluation of Biochemical Properties, Antioxidant Activities and Phenolic Content of Two Wild-Grown Berberis Fruits: Berberis nummularia and Berberis atrocarpa"

_foods, 2022, doi:10.3390/foods11172569_

Round 1
Reviewer 1 Report
Evaluation of biochemical properties, antioxidant activities and phenolic content of two wild-grown Berberis fruits: Berberis nummularia and B. atrocarpa
Topic of manuscript is interesting and valuable and is within the thematic scope of the journal. The methods used to analyze the composition and antioxidative potential of Berberis fruits are appropriate an. However there are some doubts and weaknesses which should be clarified and revised during revision process. My general and specific comments are given below:
General comments:
- The hypothesis of presented study is not properly explained in Introduction part. There were some important studies conducted which were devoted on study composition and antioxidant properties of Berberis fruits. I am not agree with this sentence in Lines: 61-63 “However, to the best of our knowledge, there are insufficient literature data for the biochemical composition and antioxidant activity of fruits of this genus in northwest China”.
Some important studies were not included in Introduction part. Thus I recommend inserting the items below during explanation hypothesis:
- Composition and antioxidant activity of the anthocyanins of the fruit of Berberis heteropoda Schrenk, Sun et al. 2014, https://doi.org/10.3390/molecules191119078
- Antioxidant, antimicrobial activities and fatty acid compositions of wild Berberis spp. by different techniques combined with chemometrics (PCA and HCA) Betül Gıdık, Molecules, 2021, 26, 7448. https://doi.org/10.3390/molecules26247448
- Phytochemicals and Traditional Use of Two Southernmost Chilean Berry Fruits: Murta (Ugni molinae Turcz) and Calafate (Berberis buxifolia Lam.) Foods, 2020, 9, 54; doi:10.3390/foods9010054
- Also in hypothesis please underline what new knowledge brings your study considering the current state of knowledge
- In introduction parts should be also mentioned about composition, antioxidant properties of other wild fruits (beeries) obtained from trees grown in natural sites (rowan fruit, Taxus baccata fruits, Hippophaë rhamnoides (sea buckthorn) and Amelanchier alnifolia (Saskatoon berries).
- I think the aim needs reformulation, please consider for what purpose should be investigated composition and antioxidative potential of Berberis fruits.
- Results of study previous Authors recommended above should be inserted in Results and Discussion part. This section is not sufficiently discussed. Please revise.
- You find differences in contents of compounds between two Berberies species. Please explain the possible use of such information (in conclusions, in abstract)
Specific comments:
- From 2.1 part please separate information about sample collection and chemicals (maybe in other new part). Also please include information about collection of samples (cultivated plantations or natural habitats).
- Lines 83-84 please insert more information about methodology of analysis of proximate composition (lipids and fiber).
- In Lines 89-96 please insert information about standards.
- Please be careful on using acronyms. For example TPC (line 131) was not previously explained.
- Line 182-184, please verify appropriateness of using term “soluble sugar and its component”.
- Line 184, I do not understand this sentence (every food source contain).
- Part 3.1: Please compare the content of proteins with its content in other wild fruits (see General Comments). Will these fruits be a good source of protein in human nutrition ?
- The results of analysis fiber content were not presented in the manuscript please supply. What does it mean RFC (please not use acronym in Table is difficult to follow).
- Table 1 is not properly and nice presented, please correct.
- Part 3.3. is not sufficiently discussed please supply more discussion.
- Total phenolic content and antioxidant properties of studied fruits should be also compared with other wild fruits not only with such properties of Berberis fruits (parts 3.4 and 3.5) please supply and revise.
- Conclusion: please delete first redundant sentence in Lines 325-327. Also reformulate conclusion pointing advantages in composition and antioxidant properties. The second sentence is too general, please indicate which element is characteristic for studied berries.
Author Response
Dear reviewer:
General comments:
Point 1: The hypothesis of presented study is not properly explained in Introduction part. There were some important studies conducted which were devoted on study composition and antioxidant properties of Berberis fruits. I am not agree with this sentence in Lines: 61-63 “However, to the best of our knowledge, there are insufficient literature data for the biochemical composition and antioxidant activity of fruits of this genus in northwest China”.
Some important studies were not included in Introduction part. Thus I recommend inserting the items below during explanation hypothesis:
- Composition and antioxidant activity of the anthocyanins of the fruit of Berberis heteropoda Schrenk, Sun et al. 2014, https://doi.org/10.3390/molecules191119078
- Antioxidant, antimicrobial activities and fatty acid compositions of wild Berberis spp. by different techniques combined with chemometrics (PCA and HCA) Betül Gıdık, Molecules, 2021, 26, 7448. https://doi.org/10.3390/molecules26247448
- Phytochemicals and Traditional Use of Two Southernmost Chilean Berry Fruits: Murta (Ugni molinae Turcz) and Calafate (Berberis buxifolia Lam.) Foods, 2020, 9, 54; doi:10.3390/foods9010054
Response: Thanks for your suggestion. The hypothesis of presented study in Introduction partis explained again. This sentence in Lines: 61-63 “However, to the best of our knowledge, there are insufficient literature data for the biochemical composition and antioxidant activity of fruits of this genus in northwest China” is modified. The important studies you mentioned were inserted in Introduction part. The introduction part modified as follows:
The nutritional content and medicinal values of wild edible fruits getting increasing value due to their attractive sensory properties as well as a global trend in finding new fruits with potential health benefits (Gıdık, 2021). Numerous studies have reported that berberis as a rich source of nutritional and phytocompounds and exhibit antioxidant (Sun et al., 2022; Gıdık, 2021). Both fruits of murta and calafate (Berberis buxifolia Lam.) grow in the wild of the Patagonia can be sources of anthocyanins with nutraceutical potential, shows a content in ascorbic acid of 210 mg/100 g FW (Arancibia-Avila et al., 2012) and 74.0 mg/100 g FW, respectively (Ruiz et al., 2010). The content of lipids and crude protein in Hippophaë rhamnoides (sea buckthorn) was 21±0.1% and 17±0.4% DWP (Dienaite et al., 2020). The antioxidant capacity of ethanol extracts of H. rhamnoides fruits as 294.1±6.53 µM TE/g (ORAC), 268.5±7.10 µM TE/g (ABTS), 102.3±4.31 (DPPH). H. rhamnoides fruits as a rich source of dietary antioxidants belonging mainly to the class of phenolic compounds, primarily proanthocyanidins, gallocatechins and flavonol glycosides. (Dienaite et al., 2020). Gıdık (2012) showed that among the cultured and Hybrid plant of B. crataegina DC., B. integerrima Bunge, and wild B. vulgaris L., the last one had the highest FRAP (621.02±25 µmol/g) values. The highest content of phenolic substances of wild Saskatoon berry (Amelanchier alnifolia Nutt.) was 3.80 g GA/kg of fresh mass) and the highest antioxidant capacity was 5.05 g of ascorbic acid equivalent (ABTS) (Rop et al., 2012). Among antioxidants, phenolic compounds as flavonoids, phenolic acids and anthocyanins have antitumor, antidiabetic and immunomodulatory effects (sun, 2014; Garcia-Diaz et al., 2019; Fredes et al., 2020; Sun et al., 2022).Therefore, in recent decades, the wild edible berberry and their extracts growing in different regions of the world widely used as a source of natural antioxidants in food and pharmaceutical industries. With these, there is an increasing demand of various products derived from berberis fruit, which led to research interests in determine the nutritional composition, biological activities and various potentials of wild edible berberis fuirts (Di et al., 2010).
Northwest China is one of the main sources of genus Berberis. Berberis nummularia Bunge and B. atrocarpa Schneid. are the two wild species distributed on valley at altitudes of 550-1100 m, in Xinjiang, Northwest China (Turahun and Muhammad, 2013). The fruit of B. nummularia and B. atrocarpa have oval and ovoid shape and contains black elliptic seeds in a red and black pulp, respectively (Li et al., 2020), which are culturally significant tree due to its traditional uses in medicine, the fresh fruits has been used as an appetizer in making juices and sauces (Turahun and Muhammad, 2006; Sun et al., 2022). For medical purposes, some studies have been carried out on the roots and fruit of B. nummularia, and B. atrocarpa. Berberine, and Vitamin C and flavonoinds are known to be isolated from the fruit, root, and peel of B. nummularia and B. atrocarpa (Di et al., 2010; sun, 2015). However, to the best of our knowledge, there are few literature data for the biochemical composition and antioxidant activity of fruits of this two species in northwest China. Therefore, the objective of this study was to assess the biochemical characterization, and evaluate the antioxidant activities of the wild edible fruits of B. nummularia and B. atrocarpa fruits for the development of functional food and nutraceuticals.
In this study, the nutrition component, mineral substance and organic acid compositions of fruits of wild Berberis nummularia Bunge and Berberis atrocarpa Schneid. were determined. With this, total flavonoid content, total phenolic content, and antioxidant (DPPH, FRAP, ABTS and ORAC) capacities were determined in order to gain an indication about the bioactive content of the samples.
Point 2: Also in hypothesis please underline what new knowledge brings your study considering the current state of knowledge.
Response: Thanks for your suggestion.
In this study, the nutrition component, mineral substance and organic acid compositions of fruits of wild Berberis nummularia Bunge and Berberis atrocarpa Schneid. were determined. With this, total flavonoid content, total phenolic content, and antioxidant (DPPH, FRAP, ABTS and ORAC) capacities were determined in order to gain an indication about the bioactive content of the samples.
Point 3: In introduction parts should be also mentioned about composition, antioxidant properties of other wild fruits (beeries) obtained from trees grown in natural sites (rowan fruit, Taxus baccata fruits, Hippophaë rhamnoides (sea buckthorn) and Amelanchier alnifolia (Saskatoon berries).
Response: Thanks for your suggestion. In introduction parts composition, antioxidant properties of other wild fruits as Hippophaë rhamnoides (sea buckthorn) and murta obtained from trees grown in natural sites are mentioned as follows:
Both fruits of murta and calafate (Berberis buxifolia Lam.) grow in the wild of the Patagonia can be sources of anthocyanins with nutraceutical potential, shows a content in ascorbic acid of 210 mg/100 g FW (Arancibia-Avila et al., 2012) and 74.0 mg/100 g FW, respectively (Ruiz et al., 2010). The content of lipids and crude protein in Hippophaë rhamnoides (sea buckthorn) was 21±0.1% and 17±0.4% DWP (Dienaite et al., 2020).
Point 4: I think the aim needs reformulation, please consider for what purpose should be investigated composition and antioxidative potential of Berberis fruits.
Response: Thanks for your suggestion. The purpose of the investigated composition and antioxidative potential of Berberis fruits as follows:
In this study, the nutrition component, mineral substance and organic acid compositions of fruits of wild Berberis nummularia Bunge and Berberis atrocarpa Schneid. were determined. With this, total flavonoid content, total phenolic content, and antioxidant (DPPH, FRAP, ABTS and ORAC) capacities were determined in order to gain an indication about the bioactive content of the samples.
Point 5: Results of study previous Authors recommended above should be inserted in Results and Discussion part. This section is not sufficiently discussed. Please revise.
Response: Thanks for your suggestion. Results of study previous Authors recommended above are inserted in Results and Discussion part.
In result part ( in line 327-329): But our total phenolic results of B. atrocarpa is also higher than most of the studies in the literature. Rybicka et al. (2021) showed that the total phenolic content of Goji, Chokeberry, Physalis and Juniper were less than 30.12 mg GAE/g.
In result part ( in line 348-351): Rybicka et al. (2021) also reported that the total phenolic content of Chokeberry and Juniper were 7.75 mg/g and 3.23 mg/g DW, however, it was reached 13.25 mg/g DW in Physalis, which was higher than the results of our study.
In Conclusion part: Wild and medicinal plants have been used for many purposes. It is important to determine the biochemical properties and antioxidant capacity of these plants when consumed as food. In this paper, for the first time, approximates the traditional knowledge of B. nummularia and B. atrocarpa fruits with the scientific research on both species. Advances in the study of the nutritional composition of fruits and the biological activity of certain phytochemicals indicate that B. nummularia and B. atrocarpa are promising sources of natural antioxidants, with nutraceutical potential. From the nutritional point of view, B. nummularia and B. atrocarpa fruits appears to be a good source potassium element. The crude fiber content is relatively high, The soluble sugar and energy value of B. nummularia was higher than that B. atrocarpa.
It is found that the TPC, TFC, as well as potentially antioxidant properties of B. atrocarpa has significant predominance. Despite the highly nutritious properties of the two wild fruits and their wide range of potential uses, they are not receiving enough attention and not properly recognized and assessed. When B. nummularia and B. atrocarpa fruits important biochemical components are considered, it can be deduced that these fruits can be used in the food industry (food coloring and nutritional supplements) as a health-promoting food, since they enclose higher potassium and calcium source, malic acid content and higher antioxidant capacity.
Point 6: You find differences in contents of compounds between two Berberies species. Please explain the possible use of such information (in conclusions, in abstract).
Response: Thanks for your suggestion. The possible use of the two species explained in the discussion and in the abstract.
In the discussion part: Advances in the study of the biological activity of certain phytochemicals indicate that B. nummularia and B. atrocarpa also are a promising sources of natural antioxidants, with nutraceutical potential. Despite the highly nutritious properties of the two wild fruits and their wide range of potential uses, they are not receiving enough attention and not properly recognized and assessed. When B. nummularia and B. atrocarpa fruits important biochemical components are considered, it can be deduced that these fruits can be used in the food industry (food coloring and nutritional supplements) as a health-promoting food and also can be consuming the fruits of the two species to weight control and reducing blood glucose or cholesterol.
In the introduction part: The fruits of the two species can be used in food coloring and nutritional supplements and also consuming the fruits to weight control and reducing blood glucose or cholesterol.
Specific comments:
Point 1: From 2.1 part please separate information about sample collection and chemicals (maybe in other new part). Also please include information about collection of samples (cultivated plantations or natural habitats).
Response: Thanks for your suggestion. 2.1. separated the information about sample collection and chemicals. Also inserted information about collection of samples ( natural habitats) as follows:
Fruits of B. nummularia and B. atrocarpa obtained from 30 different trees were collected from natural habitant in Aksu (XJBI 10020070) (41°29′85″N, 79°98′14″E) and Yili (KUN 1448344) (43°32′55″N, 84°03′46″E), Xinjiang, respectively, in October 2021 (Figure 1) and identified by professor Duan, S.M., from Xinjiang institute of ecology and geography, Chinese Academy of Sciences. The fruits were dried at room temperature and then were frozen at -80 °C until further analysis.
The organic acid standards (tartaric acid, malic acid, acetic acid, citric acid and fumaric acid) were purchased from Sigma Company of the US. Gallic acid, trolox and ascorbic acid were obtained from Yuanye Biotechnology Ltd (Shanghai, China). Methanol, ethanol and other chemicals of analytic grade were purchased from Sinopharm Chemical Reagent Co. (Shanghai, China).
Point 2: Lines 83-84 please insert more information about methodology of analysis of proximate composition (lipids and fiber).
Response: Thanks for your suggestion. The more information about methodology of analysis of proximate composition (lipids and fiber) were inserted as follows:
The ash by incineration of 3 g of each sample in a muffle furnace at 600 ℃ for 6 hours. Crude fat was extracted with n-hexane in a soxhlet extractor. Total dietary fibre content was determined with using of the Megazyme International total dietary fibre assay (adopted from AOAC method 985.29)[15].
Point 3: In Lines 89-96 please insert information about standards.
Response: Thanks for your suggestion. The information about standards is inserted as follows:
The total soluble sugar components were carried out using the method of Bouhlali et al., (2017) (Bouhlali et al., 2017), by HPLC (Agilent 1200) equipped with a refractive index detector (RID-10) (model 1200 series), a Rheodyne 7725 injector of 20 μL loop volume and a column (120A, 250 mm×4.6 mm, 3um (Athena NH2) maintained in 40 ℃. The mobile phase was a mixture of DI acetonitrile: water=(75:25 v/v). The flow rate was kept at 1.0 mL/min.
A calibration curve was prepared using fructose and glucose standard solution (0.130625, 0.26125, 0.5225, 1.045 and 2.09 mg/mL with R2=0.9984) and (0.13375, 0.2675, 0.535, 1.07 and 1.605 mg/mL with R2=0.9965), respectively.
Point 4: Please be careful on using acronyms. For example TPC (line 131) was not previously explained.
Response: Thanks for your suggestion. Total Phenolic Content is added in front of TPC.
Point 5: Line 182-184, please verify appropriateness of using term “soluble sugar and its component”.
Response: Thanks for your suggestion. The sentence corrected as “The composition including the fat, protein, fiber, ash, soluble sugar (fructose and glucose), titratable acid, carbohydrates, and energy value of berberis was showed in Table 1.”
Point 6: Line 184, I do not understand this sentence (every food source contain).
Response: Thanks for your suggestion. The line 184-185 corrected as follows:
3.1 Proximate Composition of Fruits of B.nummularia and B.atrocarpa
The composition including the fat, protein, fiber, ash, soluble sugar (fructose and glucose), titratable acid, carbohydrates, and energy value of berberis was showed in Table 1.
Point 7: Part 3.1: Please compare the content of proteins with its content in other wild fruits (see General Comments). Will these fruits be a good source of protein in human nutrition ?
Response: Thanks for your suggestion. The soluble protein content of the two berberis is higher than that the fruits of B. microphylla[10], in which, the soluble protein content was only 1±0.1%. However, the protein content of the two species were lower than that in Hippophaë rhamnoides (sea buckthorn), in which, the protein content was 17±0.4% DWP[5], these fruits be a good source of protein in human nutrition.
Point 8: The results of analysis fiber content were not presented in the manuscript please supply. What does it mean RFC (please not use acronym in Table is difficult to follow).
Response: Thanks for your suggestion. The results of analysis fiber content were presented in the manuscript as follows:
Dietary fiber is increasingly viewed as an essential aspect of good nutrition. The fiber content is up to 16% for both B. nummularia and B. atrocarpa, which is in line with the existing data of other berberis fruit[27]. Researchers suggest that the consumption of dried fruits rich in dietary fiber is beneficial for individuals such as weight control and reducing blood glucose or cholesterol[28], it may be considered as an advantage due to increased satiety and intestinal regulation[29]. The content of carbohydrates was about 56% for the two fruits and it resulted mainly from the high content of dietary fiber. The energy value was between 330.86 kcal for B. nummularia and 314.41 kcal/100 g for B. atrocarpa.
Point 9: Table 1 is not properly and nice presented, please correct.
Response: Thanks for your suggestion. Table 1 is presented as follows:
|
fruits |
B. nummularia |
B. atrocarpa |
|
Protein [g/100 g d.m.1] |
2.593±0.091a |
4±0.1b |
|
Fat [g/100 g d.m.1] |
6.77±0.42a |
5±0.1b |
|
RFC [g/100 g d.m.2] |
16.4±1.0a |
18±1.4a |
|
Ash [g/100 g d.m.1] |
1±0.1a |
1±0.0a |
|
Soluble sugar [g/100g d.m.1] |
23±0.6a |
12±1.46b |
|
Fructose (mg/g d.m.1) |
36±0.2a |
6±0.1b |
|
Glucose (mg/g d.m.1) |
29±0.1a |
10±0.0b |
|
Titratable acid(mg/g d.m.1) |
18±2.5a |
14±1.26b |
|
carbohydrates [g/100 g d.m.3] |
57±1.8 |
56±1.8 |
|
Energy value[Kcal/100g d.m.4] |
330.86a |
314.41b |
Point 10: Part 3.3 is not sufficiently discussed please supply more discussion.
Response: Thanks for your suggestion. Part 3.3 is discussed more.
Organic acids are widely used as preservatives, antioxidants, acidulants, and drug absorption modifiers[33], which influence fruit qualities such as fruit coloration and ripeness. Organic acids are different in various kinds of fruits. For example, citric acid is the major organic acid in citrus[34] and wild apples[35], while malic acid is the predominant organic acid in cultivated apple[36], pear[37]. Glew et al. (2003)[38] showed a type of wild fruit called medlar (Mespilus germanica L.) was reported to contain around 4 g/kg fresh weight of citric acid and malic acid, it is likely due to the genetic effects of parental characters.
Our results showed there were tartaric acid, malic acid, citric acid, acetic acid and fumaric acid in B. nummularia and B. atrocarpa fruits (Figure 2). The organic acid content ranged from 0.48 to 118.8 mg/g. There were significant differences (P<0.001) in the content of total organic acid between the two fruits (Figure 2). Malic acid was the main organic acid for the two fruits, it was significantly higher in B. nummularia (119±0.1 mg/g) than that B. atrocarpa (84±0.1 mg/g). The content of citric acid and tartaric acid varied from 0.48 to 4.23 mg/g. The fumaric acid was present at low concentrations in both two fruits.
Point 11: Total phenolic content and antioxidant properties of studied fruits should be also compared with other wild fruits not only with such properties of Berberis fruits (parts 3.4 and 3.5) please supply and revise.
Response: Thanks for your suggestion. Total phenolic content and antioxidant properties of studied fruits is compared with other wild fruits as follows:
But our total phenolic results of B. atrocarpa is also higher than most of the studies in the literature. Rybicka et al. (2021)[27] showed that the total phenolic content of Goji, Chokeberry, Physalis and Juniper were less than 30.12 mg GAE/g.
Rybicka et al. (2021)[27] also reported that the total phenolic content of Chokeberry and Juniper were 7.75 mg/g and 3.23 mg/g DW, however, it was reached 13.25 mg/g DW in Physalis, which was higher than the results of our study.
Point 12: Conclusion: please delete first redundant sentence in Lines 325-327. Also reformulate conclusion pointing advantages in composition and antioxidant properties. The second sentence is too general, please indicate which element is characteristic for studied berries.
Response: Thanks for your suggestion. The first sentence in Lines 325-327 is deleted. The conclusion is reformulated as follows:
Wild and medicinal plants have been used for many purposes. It is important to determine the biochemical properties and antioxidant capacity of these plants when consumed as food. In this paper, for the first time, approximates the traditional knowledge of B. nummularia and B. atrocarpa fruits with the scientific research on both species. Advances in the study of the nutritional composition of fruits and the biological activity of certain phytochemicals indicate that B. nummularia and B. atrocarpa are promising sources of natural antioxidants, with nutraceutical potential. From the nutritional point of view, B. nummularia and B. atrocarpa fruits appears to be a good source potassium element.
Reviewer 2 Report
The article has an interesting topic. However, the presentation of the results is poor due to many linguistic and stylistic errors. It is difficult to focus on the merits wondering what the Authors meant. Unfortunately, the text must be corrected by a person who knows English very well or a native speaker. Below, I list the most important elements that need to be improved.
Title: Do not use abbreviations in the title. Berberis atrocarpa should be written in its full name.
Please put full stops after the titles of tables and figures (where they are missing).
In the names of B. atrocarpa and B. nummularia, please use a space after "B.". This should be corrected throughout the text of the article.
Line 21: The word "content" is missing from the sentence "The results indicated that no meaningful differences (P> 0.05) between the ash (0.97 ± 0.08 and 0.97 ± 0.04 g / 100 g), fiber (16.4 ± 0.96 and 17.6 ± 1.42) and carbohydrate ( 56.85 ± 1.77 and 55.68 ± 1.82 g / 100 g) content, respectively, in dry fruits of B.nummularia and B.atrocarpa.".
Lines 23, 87: "titratable acidity" not "titratable acid".
Lines 49, 50: "berberry" not "barberry".
Line 72: "were" not "ware".
Line 82: In the title of the chapter 2.2 "Proximate Composition Analysis of B. nummularia and B. atrocarpa fruit" please use the plural "fruits".
Line 88: The calculation formula appears without any prior comment.
Line 97: In the title of the chapter 2.3 "Organic Acid Determination" please use the plural "acids".
Line 101: The sentence "Samples were further centrifuged at 10000 r/min ..." contains the unit of "r/min". This abbreviation is not used. The abbreviation "rpm" is generally accepted. In addition, the given value should be converted into the unit "g".
Line 102: Grammar error - "The supernatant were collected". It should be "The supernatants were collected".
Line 103: "equipped" not "quipped".
Chapter 2.3: Please provide the type, model and name of the manufacturer of the HPLC system and the chromatography column as well as their full details (city, state - applicable eg USA, country).
Line 113: "powdered sample" not "sample powder".
Line 123: Before "5g", remove the dot/full stop.
Line 131: Error in method name. It should be "Folin" not "Foline".
Chapters 2.5.1, 2.5.2, 2.6.1, 2.6.2: Please enter the wavelengths at which the absorbances were measured.
Chapter 3.3: Please write the title "The Organic Acid Contents" in the plural - The Organic Acids Contents.
These are the most important points. Others may appear after the Authors' revision.
Author Response
Dear reviewer:
Point 1: Title: Do not use abbreviations in the title. Berberis atrocarpa should be written in its full name.
Response 1: Thank you very much for your suggestion. Berberis atrocarpa is written in its full name in the title.
Point 2: Please put full stops after the titles of tables and figures (where they are missing).
Response 2: Thanks for your suggestion. Full stops after the titles of tables and figures were added.
Point 3: In the names of B. atrocarpa and B. nummularia, please use a space after "B.". This should be corrected throughout the text of the article.
Response 3: Thanks for your suggestion. All the scientific name entries, include a space between the period and the specific epithet.
Point 4: Line 21: The word "content" is missing from the sentence "The results indicated that no meaningful differences (P> 0.05) between the ash (0.97 ± 0.08 and 0.97 ± 0.04 g / 100 g), fiber (16.4 ± 0.96 and 17.6 ± 1.42) and carbohydrate ( 56.85 ± 1.77 and 55.68 ± 1.82 g / 100 g) content, respectively, in dry fruits of B.nummularia and B.atrocarpa.".
Response 4: Thanks for your suggestion. The word "content" is added to the sentence.
Point 5: Lines 23, 87: "titratable acidity" not "titratable acid".
Response 5: Thanks for your suggestion. The "titratable acid" corrected as "titratable acidity" in line 23 and line 87.
Point 6: Lines 49, 50: "berberry" not "barberry".
Response 6: Thanks for your correction. "barberry" in Lines 49, 50 is corrected as "berberry".
Point 7: Line 72: "were" not "ware".
Response 7: Thanks for your correction. "ware" in Line 72 is corrected as "were".
Point 8: Line 82: In the title of the chapter 2.2 "Proximate Composition Analysis of B. nummularia and B. atrocarpa fruit" please use the plural "fruits".
Response 8: Thanks for your suggestion. In the title (in line 82) of the chapter 2.2 "Proximate Composition Analysis of B. nummularia and B. atrocarpa fruit" is used the plural "fruits" and this title is corrected as “Proximate Composition Analysis of B. nummularia and B. atrocarpa fruits”.
Point 9: Line 88: The calculation formula appears without any prior comment.
Response 9: Thanks for your comments. Energy was expressed as kcal/100g, using the following formula:
Point 10: Line 97: In the title of the chapter 2.3 "Organic Acid Determination" please use the plural "acids".
Response: Thanks for your suggestion. The title in line 97 of the chapter 2.3 "Organic Acid Determination" is used the plural "acids".
Point 11:Line 101: The sentence "Samples were further centrifuged at 10000 r/min ..." contains the unit of "r/min". This abbreviation is not used. The abbreviation "rpm" is generally accepted. In addition, the given value should be converted into the unit "g".
Response:Thanks for your suggestion. The given value (10000 rpm/min) is converted into the unit "g" as 9050g/min.
Point 12: Line 102: Grammar error- "The supernatant were collected". It should be "The supernatants were collected".
Response: Thanks for your correction.“The supernatant were collected" is corrected as "The supernatants were collected".
Point 13: Line 103: "equipped" not "quipped".
Response: Thanks for your correction. "quipped" is corrected as "equipped".
Point 14: Chapter 2.3: Please provide the type, model and name of the manufacturer of the HPLC system and the chromatography column as well as their full details (city, state - applicable eg USA, country).
Response: Thanks for your suggestion. HPLC (Agilent 1260, US).
Point 15: Line 113: "powdered sample" not "sample powder".
Response: Thanks for your correction. sample powder" is corrected as "powdered sample".
Point 16: Line 123: Before "5g", remove the dot/full stop.
Response: Thanks. It is removed.
Point 17: Line 131: Error in method name. It should be "Folin" not "Foline".
Response: Thanks for your correction. The method name is corrected as "Folin".
Point 18: Chapters 2.5.1, 2.5.2, 2.6.1, 2.6.2: Please enter the wavelengths at which the absorbances were measured.
Response: Thanks for your correction. The wavelengths at which the absorbances of the samples were added.
Point 19: Chapter 3.3: Please write the title "The Organic Acid Contents" in the plural - The Organic Acids Contents.
Response: Thanks for your suggestion. The title "The Organic Acid Contents" is written in the plural.
Reviewer 3 Report
In general the manuscript is well written, easy to understand. In the attached file, there are correction suggestions, highlighted in yellow. Most of it corresponds to the handling of significant figures. In all scientific name entries, you must include a space between the period and the specific epithet.
The word polyphenol should also be deleted, since the technique used does not discriminate between phenols and polyphenols. (You should refer to Edwin Haslam's definition of plant polyphenols or tannins.
Author Response
Dear reviewer:
Point 1: (Comments and Suggestions) In general the manuscript is well written, easy to understand. In the attached file, there are correction suggestions, highlighted in yellow. Most of it corresponds to the handling of significant figures. In all scientific name entries, you must include a space between the period and the specific epithet.
Response 1: Thank you very much for your comments and suggestion.
- All the scientific name entries, include a space between the period and the specific epithet.
- All the significant figures was handled one by one.
Point 2: please, provide the voucher number and Herbarium name.
Response 2: Voucher number of B. nummularia and B. atrocarpa are XJBI 10020070 and KUN 1448344
Point 3: The word “A” in line 90 need to alter to “Å”.
Response 3: Thanks for your suggestion. It was altered.
Point 4: The sentence in line 103 -104 “The organic acids were detected by a HPLC quipped with variable wavelength vacuum chromatograph and high-efficiency UV detector” need to modify as “the organic acids were detected by an HPLC equipped with variable wavelength vacuum chromatograph and high-efficiency UV detector”.
Response 4: Thanks for your suggestion. The sentence was modified.
Point 5: The words ”total polyphenol content” in line 122 need to be deleted.
Response 5: Thanks for your suggestion. The words”total polyphenol content” is deleted.
Point 6: “Foline-Ciocalteu colorimetric” in line 131 is need to corrected as “Folin-Ciocalteu colorimetric”.
Response 6: Thanks for your suggestion. It was corrected.
Point 7: The words “The Ferric Reducing Ability of Plasma (FRAP)”need to corrected as “ferric reducing antioxidant power assay
Response 7: Thanks for your suggestion. It was corrected.
Point 8: line 155-”•” and “stock solution” need to separate.
Response 8: Thank you. It was separated.
Point 9: figuere 3 (line 290)、line299、line302- “polyphenol content” need to alter as “phenol content”.
Response 9: Thank you. The “polyphenol content” was altered as “phenol content”.
Point 10: line 328- “rude” need to corrected as “crude”.
Response 10: Thank you. It was corrected.
Point 11: line 329- need to delete the words “the two”.
Response 11: Thank you. It was corrected.
Round 2
Reviewer 1 Report
Dear Authors,
I found that manuscript was sufficiently revised. I also appreciated responses on all comments.
Author Response
Thank you. English language and style are checked and modified.